# The Inclusion of Resilience as an Element of the Sustainable Dimension in the LOMLOE Curriculum in a European Framework

Elisa Gavari-Starkie [1,*] , Josep Pastrana-Huguet [2,3] , Inmaculada Navarro-González [4] and Patricia-Teresa Espinosa-Gutiérrez [3]

1   Department of History of Education and Comparative Education, National Distance Education University (UNED), 28015 Madrid, Spain
2   Consell Insular de Menorca, Balearic Islands, 07769 Menorca, Spain; joseppastrana@gmail.com
3   International Doctoral School of the UNED (EIDUNED), 28040 Madrid, Spain; pt.espinosagutierrez@gmail.com
4   Department of Research Methods and Diagnosis in Education, National Distance Education University (UNED), 28015 Madrid, Spain; mainavarro@edu.uned.es
*   Correspondence: egavari@edu.uned.es

**Abstract:** This article provides the research community with a conceptual framework from a historical perspective of the impulse of education sustainability in the official international literature. In addition, the United Nations International Conferences held on Japanese territory in order to foster education for risk reduction and for training resilient individuals and communities are analyzed. The study of the content of both approaches, education for sustainability and education for risk reduction, constitute an innovative approach especially relevant after the pandemic caused by the COVID-19 crisis. The article advances with a historical analysis of the use of the concept of resilience in the European Institutions' official documents. Our findings show that it is particular after 2015 when resilience is linked to sustainability. Before this, the European approach was mostly linked to food crises and emergencies. The article offers a synthesis of the global and European approaches in tables so that we can compare the progress in the United Nations discourse and the European Union one. In this conceptual framework, we offer a contribution to the debate for European national education systems. In particular, we offer contribute to the debate of the Organic Law LOMLOE approved in Spain in 2020, in which education for sustainability is strongly considered but not so much resilience education. The article intends to contribute to the inclusion of resilience as an element of the curriculum linked to the education for sustainability.

**Keywords:** education for sustainability; education for disaster risk reduction; education systems reforms; European education; resilience

## 1. Introduction

The causes of climate change and its effects are intrinsically linked to human actions, and therefore, mitigating these effects requires education action. The research on the topic can help to discuss, visualize and share new alternatives to generate collective action [1]. Several researchers such as Kagawa and Selby [2]; Anderson [3]; Monroe, Plate, Oxarart et al. [4]; and Meira [5] argue that education can show people have a critical role to play in redefining their lifestyles to address current sustainability issues facing humanity. Education offers an opportunity to build a sustainable world, and in this framework, learning processes allow building resilient skills. However, education policies that foster individual and social resilience through sustainable development remains a challenge.

The present article offers a historic evolution of the international official literature concerning education for sustainability. In this framework, we analyze the contribution of education for training resilient individuals and communities. For this purpose, we

also offer a brief summary of the United Nations' recommendations, which are a result of the three World Conferences in Disaster Risk Reduction that have been held on Japanese territory since the 1990s. In addition, we analyze the European official strategies and recommendations in sustainability and resilience.

In the framework of the international conceptual analysis provided in the first part of the article, we analyze education for sustainability in a national case within the European Union. In particular, we study the Spanish education system analyzing the inclusion of education for sustainability from a historic point of view. The results of the research show that the Spanish education systems have included sustainability but in a very limited way. However, the last Organic Law in 2020 LOMLOE has incorporated sustainability as an axis of the Spanish education system. Our claim in the article is that following the international recommendations on resilience and education for risk education are contents that must be included and linked to education for sustainability. The Royal Decrees that will further the Organic Law constitute an opportunity for linking education for resilience and sustainability.

## 2. Materials and Methods

The methodology adopted to address this research is based, firstly, on the selection of primary sources published by international organizations such as the United Nations and the European Community Institutions between 1960 and 2021in which educational sustainability and resilience are addressed. The reports have been chosen from relevant international databases. The selection criteria have been based on the international nature and global scope of the terms such as sustainability, resilience and education for disaster reduction. In this framework, a study of the scientific research articles related to the subject study has been carried out through various bibliographic databases and searches of the tables of contents of relevant high-impact international journals.

The next step has been the to choose a European national case study to analyze the incorporation of the international recommendations and guidelines into a European educational system. We have chosen the Spanish one because in 2020 the Organic Law LOMLOE was approved incorporating sustainability as the educational axis of the law. Another reason to choose this education system is that the Royal Decrees that widen the regulations have not yet been drafted. Our claim is the crisis of the still going on pandemic caused by COVID-19 is an opportunity to update the education system and include resilience linked to sustainability as an element of the curriculum.

## 3. International Contributions to the Conceptual Evolution of Environmental Education to Sustainability

Concern for the environment in the international framework has evolved from the second half of the 20th century. We can place the starting point of modern environmentalism in the 1960s, when as pollution increases, the notable deterioration of the environment is perceived as a consequence, which triggers an awareness of its detriment. In this context, the Club of Rome was organized in 1968, in which academics, politicians, researchers and scientists shared their concern for ecology and the environmental changes that were taking place. This concern was translated into the search for knowledge, awareness and action to stop environmental modifications [6]. This new awareness allowed the genesis of the first ecological education programs, which laid the foundations for subsequent Environmental Education.

In the 1970s, the concern for the conservation and protection of natural resources, fauna and flora materialized [6], laying the foundations for environmental education. Specifically, the 1975 Belgrade Charter [7] sets out the situation of "a growing deterioration of the physical environment" and defines the guiding principles of Environmental Education programs. Another milestone in environmental education is the Tbilisi Conference (1977) [8] organized by UNESCO in collaboration with the United Nations Environment Program (UNEP), being the first Intergovernmental Conference on Environmental Educa-

tion in which the starting point of the International Program of Environmental Education was established.

The 1980s are defined by the Environmental Education and Training Conference in Moscow (1987) [9], which represents a substantial change, since a new definition of Environmental Education is defended as a "permanent process in which individuals and communities acquire awareness of their environment and they learn the knowledge, values, skills, experiences and also the determination that enables them to act (...) in solving environmental problems (...)". This new approach to Environmental Education is expanded with the Report of the Nations known as Brudtland [10] on "Our Common Future" that bets, for the first time, on the concept of Sustainable Development. The fundamental contribution of this concept is to refer to sustainability as a common concern to achieve a balance between development and the environment, in a future in which the latter would be seriously threatened.

The environmental crisis with which the 1990s began, caused by factors such as deforestation, climate change, wars, famines, etc., raises with more force the need to reconcile economic development with respect for the environment [11]. An issue that reaches a substantial advance at the end of the 20th century, linking the definition of sustainability to ethics and a new vision of human development. As stated by Machín et al. [12], it is then when the human factor is required as an essential element of sustainable development and on which the pillars of ethics for sustainability are based.

The United Nations Conference held in 1992 on Environment and Development in Rio de Janeiro [13] meant the promotion of the concept of Sustainable Development [6], which is derived from two fundamental principles. First, Principle 1 states that "human beings are at the center of concerns related to sustainable development. They have the right to a healthy and productive life in harmony with nature". Second, Principle 27 states that "States and individuals shall cooperate in good faith and in a spirit of solidarity in the application of the principles enshrined in this Declaration and in the further development of international law in the field of sustainable development".

In this new framework to promote education for sustainability, Agenda 21 [13] is approved, in which Chapter 36, entitled Education, Training and awareness of Agenda 21, stands out. In particular, in Section 36.3, "it must be recognized that education—including academic teaching—public awareness and training, configure a process that allows human beings and societies to fully develop their latent capacity".

The end of the millennium supposes the definition of new proposals that materialize in the document the Millennium Development Goals (MDG) [14] that includes 8 goals and 20 goals, among which 2 stand out that refer to education. In the first place, Objective 2, entitled Achieve universal primary education and secondly, and Objective 7, entitled Guarantee environmental sustainability, of which the seventh goal includes, "Incorporate the principles of sustainable development in national policies and programs and reduce the loss of environmental resources", all the objectives being aimed at achieving development values for the year 2015.

To complete the new proposal of the MDGs, the World Summit on Sustainable Development [15] was held in Johannesburg in 2002, which led to the approval of the Political Declaration and the Plan of Implementation. The new approach places a special emphasis on the educational field, which is notoriously far from being about sustainability, clearly reaffirming itself as being for sustainable development, for sustainability. Regarding this event, we can remember Martínez et al. [16], who point out that the goal of environmental education is education for sustainability. There are several articles that refer to education in this statement. As examples, we can point out that article 18 of the Declaration [17] specifies "(...) in order to achieve development, ensure that the transfer of technology, the improvement of human resources, education and training are promoted in order to eradicate always underdevelopment".

On the other hand, in the Implementation Plan, the importance of education is strengthened, pointing out the need to promote some changes. Section 8d insists on the need to

"Promote education and outreach focused on children, as agents of behavior change", and Section 15d points out the importance of "Developing programs to sensitise the public about the importance of sustainable forms of production and consumption, in particular to young people (...), especially in developed ones through, among other things, education". In Section 43m, you can read, "Promote education to provide both men and women with information on energy sources and available technologies". In Section 111, we can see again the importance of education for sustainable development: "(...) create or strengthen scientific and educational networks for sustainable development at all levels (...)". In Section 116, it expressly cites that "Education is of critical importance to promote sustainable development", and in Section 117, it states, "Providing financial assistance and support to teaching, research, public awareness programs and development institutions in developing countries and countries with economies in transition", set out in Sections a and b, the goals to be achieved in education.

In 2005, the United Nations took another step forward and declared the Decade for Education for Sustainable Development (2005–2014) [18] in order to mobilize educational resources from a triple civic, ethical and moral orientation. Saura and Hernández [6] had the larger purpose of achieving a more sustainable world.

Finally, from the year 2015, it is committed to a concept of sustainability considered from a holistic approach or, in other words, by a holistic sustainability. The document that establishes the new approach to education for sustainability that is still in force is the one that refers to the so-called SDGs (Sustainable Development Goals) [19] with 17 goals and 169 targets and the 2030 Global Action Plan. Goal 4 of education [20] from quality is included in goal 4.7 Global Education for Sustainable Development, which states, "By 2030, ensure that all students acquire the theoretical and practical knowledge necessary to promote sustainable development, including through education for sustainable development and sustainable lifestyles..." [20].

The COVID-19 crisis and the consequences experienced from climate change in the first third of the 21st century have marked a new turning point at the global level that requires linking sustainable education with the creation of resilient people and communities. Although sustainability aspires to a persistent and equitable long-term well-being that is summarized in the dimensions of resilience [21], educators must respond to the challenge of preparing current and future generations to make decisions that mitigate the effects of global warming. Therefore, learning in sustainability and resilience is imperative to guarantee human resilience in the face of these new challenges and adapt to new circumstances [22].

Nielsen and Faber (2019) [23] argue that a learning that encompasses risk, sustainability and resilience should be based on the theoretical principles of systems thinking, through Goodman's [24] decision analysis, inquiry-based learning and transitional learning. In addition, they identify four relevant characteristics for the development of education and integration of resilience and sustainability:

- A liquid knowledge based on permanent transformation;
- A transdisciplinary education, which includes social responsibility and participation;
- The promotion of intellectual knowledge and education in civil society values.

These characteristics require a high level of plasticity in the design of a learning environment that can accommodate both the dynamic nature of the knowledge content and the dynamic engagement of society.

## 4. The Promotion of Education for Resilience from the World Conferences Organised by the United Nations on Japanese Territory

The origin of the term "resilience" comes from the Latin word "resiliere". The term resilience was first defined in 1973 in the field of ecology by Holling [25] as the ability of an ecosystem to absorb change and return to its normal stability after a temporary disturbance. As a general concept, experts agree to define it as the potential of people, a community or society, a system, infrastructures and ecosystems to face existential risks and resist or

recover from the adverse effects of a crisis. According to some authors (Klein, Nicholls, Thomalla, 2003; Paton & Johnston, 2006) [26,27], resilience is the ability of a system to quickly recover its original state ("to bounce back," "to jump back").

Even though various studies offer other definitions of resilience, all of them agree on the capacity to adapt and the strengthening after an adverse situation [28–32]. In addition, experts have sought to complement resilience with other social theories and show that research on resilience must be carried out at various levels of analysis: individual, group, and organizational or community in a wide variety of disciplines [33,34]. From the framework of disaster management, compared to the traditional approach to development, the resilience approach (Overseas Development Institute, 2013) [35] assumes the need to build capacities, adaptive and flexible planning where various sectors, levels and actors intervene, among which are those that belong to the education sector.

Some authors such as Martínez González [31] argues that resilience constitutes a metacompetence with two essential components: on the one hand, resistance to destruction and, on the other hand, the ability to forge a positive vital behaviour despite difficult circumstances allowing a person or social system to cope with difficulties adequately.

Moreover, Zabaniotou [36] argues that it is even possible to set a chronology for resilience conceptualization:

- The first generation defined resilience as "the adaptability of the individual, who is capable of handling and overcoming adversity".
- The second generation added the term "positive adjustment", favoring individuals to be stronger and more productive and, later, positive psychology emerged.
- The third generation added the notion of "transformative change".
- The fourth generation investigated the critical issues of "equitable resilience" to participate equitably in resilience practice.

From this chronological perspective, resilience is intrinsically linked to the field of Disaster Risk Reduction and its management, which has been defined in the three Word Conferences. The following international summits have defined education as essential to face the consequences of disasters.

The United Nations General Assembly declared in the late 1980s the International Decade for Natural Disaster Reduction 1990 to 1999. This framework formed the basis for the holding of the three United Nations world conferences for disaster risk reduction, all of which have been held on Japanese soil in order to reduce human losses and material damage. It is not surprising that Japan is the driving country for these summits since, due to the geographical location it occupies and the threats suffered by both natural and human agents, it has a long tradition in history in Disaster Risk Reduction (RRD). The first World Conference on RRD was held in Yokohama in 1994 and resulted in the approval of the Strategy and Plan of Action for a Safer World [37]. This text proposed, among other issues, a greater emphasis on the social sciences in fields such as research, public policy development and practical applications, highlighting the links between disaster risk reduction and sustainable development (ISDR, 1999) [38]. In addition, the Yokohama Strategy and Action Plan established a set of guidelines for action in disaster risk prevention, preparedness and mitigation based on a set of principles that highlighted the importance of risk assessment, preparedness, the capacity to prevent, reduce and mitigate disasters and early warning (International Institute for Sustainable Development) [39].

In 1999, the United Nations General Assembly adopted the International Strategy for Disaster Reduction to promote a "culture of prevention". Following this line of action, in 2005, the Second World Conference on Natural RRD [40] was held in Hyogo, and the approval of a new action plan for the period 2005–2015 was established. This plan focused on fostering a culture of disaster prevention through education, in which it was proposed to use knowledge, innovation and education to build a culture of prevention and resilience at all levels. Authors such as Pal, Von Meding and Klinmalai [41] mention that the convergence of RRD and sustainable development is explicitly recognized in the

Hyogo Framework for Action, highlighting disaster risk reduction as an essential element in sustainable development.

The Third World Conference held in Sendai for RRD in 2015 resulted in the approval of a new 15-year framework for action in which the emphasis was placed on risk management rather than disaster management. The "Sendai Framework for Disaster Risk Reduction 2015–2030" [42] identifies seven global goals for 2030. The first four propose the reduction of damages and losses of global mortality caused by disasters, of the number of people affected, of direct economic losses relative to world gross domestic product (GDP) and of damage to vital infrastructure and disruption of basic services. The next three aspire to increase the number of countries that design national and local strategies for disaster risk reduction. In addition, it establishes four priorities for action focused on understanding disaster risk, strengthening disaster risk governance to manage risk, investing in disaster risk reduction for resilience and increasing disaster preparedness to deliver an effective response and "build back better".

Furthermore, in what concerns sustainability, the Sendai Framework provides specific opportunities to achieve the SDGs through reducing disaster risk. In fact, according to the new conceptual framework for the application of the Sendai Framework 2016 [43], resilience is defined as the capacity of a system, a community or a society exposed to a threat to resist, absorb, adapt, transform and recover from its effects in a timely and efficient manner, in particular by preserving and restoring its basic structures and functions through risk management. The resilience strategy is based on four fundamental aspects:

- Creation of a favorable environment, based on institutional strengthening and risk management;
- Surveillance to safeguard, implying the implementation of information and early warning systems;
- Application of risk and vulnerability reduction measures, promoting environments for protection, prevention, mitigation and construction of livelihoods with technologies, approaches and practice;
- Preparation and response, in the face of the crises generated.

Another concept is Disaster Risk Reduction, which aims to increase resilience to natural disasters such as earthquakes, floods and cyclones, among other shocks and crises, through the identification and effective management of risks. The underlying purpose of disaster risk reduction efforts definitely influences policy, planning and activities related to how to identify, design and implement development assistance. In practice, this is achieved through the creation of a risk-based culture, establishing risk and vulnerability analysis processes, improving capacity and technology and facilitating access to risk information.

The 2030 Agenda for Sustainable Development includes several goals and targets that can help reduce disaster risk and build resilience. These include objectives related to the promotion of education for sustainable development, the construction and improvement of educational facilities and healthy life, among others (UNDRR, 2015) [42,43] (see Table 1).

**Table 1.** Evolution in the global framework of education for sustainability and resilience.

| Period of Concern | Summits and Programs of Education Sustainability | Contents | Resilience |
|---|---|---|---|
| 1950–1970 Growing concern about environmental changes. | Ecological education programs are created. | | 1957: Treaty of Rome. Inclusion of general economic cooperation (Economic resilience). |
| 1970–1980 Inevitable emergence of the concept of environmental education (EA) to help solve the problems that occur in the environment. | The Belgrade Charter (1975) | Sets out the general framework for environmental education, highlighting all the guiding principles of environmental education programs. | |

**Table 1.** *Cont.*

| Period of Concern | Summits and Programs of Education Sustainability | Contents | Resilience |
|---|---|---|---|
| | The Tbilisi Conference (1977) | The first culminating point in the first phase of the International Environmental Education Program, initiated in 1975 by UNESCO, with the cooperation of the United Nations Environment Program (UNEP). | |
| 1980s–1990s Debate around development. New appearance of the definition of Environmental Education. | The Environmental Education and Training Conference held in Moscow (1987) | Includes a new definition of EE. | The United Nations International Decade for Natural Disaster Reduction 1980–1990 |
| | The Brundtland Report (1987) | | |
| 1990–2000 Environmental crisis drives the emergence of education for sustainability. | The Rio Summit (1992) allows us the emergence of an education for sustainability. | In its Chapter 36 of Program 21 of Agenda 21 recognizes education for development. | 1992: United Nations Framework Convention on Climate Change. |
| | The International Conference on Environment and Society: Education and Public Awareness for Sustainability held in Thessaloniki (1997). | The relationship between poverty, development and the environment. | 1994: Yokohama First World Conference on Disaster Reduction. |
| 2000–2010 Emergence with the rise of the link between Environmental Education and the so-called Education for Sustainable Development (ESD). | The Millennium Development Goals (2000–2015) [9] emerge. | | 2005: Second World Conference on Natural Disaster Risk Reduction (2005–2015). |
| | The World Summit for Sustainable Development, Johannesburg (2002). | Promote sustainability and sustainable development. | EU 2009 |
| | The Decade of Education for Sustainable Development (2005–2014) emerges. | | The EU approach to resilience: learning from food crises. |
| 2010–(…) 2030 Agenda | The SDGs (Sustainable Development Goals) and the 2030 Global Action Plan emerged in September 2015. | The current conception of the concept of sustainability in education revolves around the 2030 Agenda and its 17 SDGs, appearing a holistic approach. | Third Sendai World Framework Conference for Disaster Risk Reduction (2015–2030). |
| | Target 4.7 refers to the achievement of education for sustainable development by 2030. | | The SDG document refers to resilience. |
| | | | EU Document 2016 "A common vision, joint action: a stronger Europe. Global strategy for the foreign and security policy of the European Union" 2016 European Skills Agenda. |
| Global crisis of the COVID-19 pandemic, concern about other possible future pandemics. Education for a sustainable and resilient world. | | | 2020: New Agenda for European Capacities for sustainable competitiveness, social equity and resilience. |
| | | | 2020: Europe Recovery Plan. |

Source: Information extracted from the different funds and instruments.

Currently, with the COVID-19 pandemic crisis and the increase in natural disasters (such as, volcanic eruptions) due to climate change, governments worldwide and institutions such as the UN consider it urgent to reinforce the link between resilience and sustainability, which are intrinsically connected to disasters. As Sakurai and Sato men-

tioned [44], these challenges imply actions of monitoring and evaluating models to measure the impact of education in attitudes and behavior changes.

## 5. Contributions to Education for Sustainability and Resilience from the European Guidelines and Recommendations

Since 2009, and in line with international guidelines, the European Union has also worked to promote resilience. If initially resilience was approached from the perspective of RRD associated with the geographic area of developing countries [45], in 2012 there are references to resilience to humanitarian and humanitarian issues development in the Communication of the European Commission entitled "The EU approach to resilience: learning from food crises" [46]. Shortly thereafter, in the 2013 Council Conclusions, resilience is defined as "the ability of a person, a household, a community, a country or a region to prepare, cope, adapt and recover quickly from stresses and shocks. without jeopardizing long-term development expectations".

From 2015, the EU has incorporated the recommendations of the 2030 Agenda [19,20] established for the period 2015–2030 through the document "A common vision, joint action: a stronger Europe. Global strategy for the foreign and security policy of the European Union" [47]. This document calls for the participation of civil society to achieve resilience. In particular, the document calls for the strengthening of social resilience from the field of education, culture and youth to promote pluralism, coexistence and respect. The document understands that the core of a resilient state lies in democracy, trust in institutions and sustainable development. The perspective of this approach delegates to citizens and communities the capacity to manage opportunities and risks and positions education as a key element for the development of resilience.

Another of the key documents in education for resilience development is the Europe Skills Agenda [48]. This document makes explicit reference to the enhancement of resilience through the acquisition of key competences that develop superior and more complex ones and that help people to increase their potential at work and in society. The 2016 Skills Agenda has recently been updated with the New Skills Agenda, which is made up of 12 actions for sustainable competitiveness, social equity and resilience [49] and which are related to training and professional retraining. These new actions revolve around the real-time analysis of the needs of the labour market and the skills necessary to cover them in the hands of national public employment organisms.

The Skills Agenda is accompanied by a proposal for a recommendation from the EU Council on Vocational Education and Training (VET) [50] where reference is made to the changes that Education and Vocational Training (VET) should be carried out to adapt to changes in the labour market and a society in constant evolution from proactive participation and personal development. In support of these reforms and investments in VET in terms of digitization, environmental sustainability and resilience, among other things, the European Union has several funds and instruments, namely: 'Next Generation EU' (Recovery and Resilience Mechanism, REACT-EU) [51], the European Social Fund + [52], the SURE program [53], the European Regional Development Fund [54], InvestEU [55], Horizon Europe [56], Digital Europe [57], the Mechanism for a Just Transition [58] and the European Agricultural Fund for Rural Development [59] (see Table 2).

**Table 2.** EU funds and instruments to promote sustainability and resilience in public policies.

| EU Tools | Contents |
|---|---|
| European Social Fund Plus (European Commission, 2018) [54] | Highest levels of employment<br>• Fair social protection;<br>• Development of a skilled and resilient workforce, prepared for the transition to a green and digital economy. |
| Horizon Europe (European Commission, 2019) [56] | Excellent science<br>• Digital challenges and European industrial competitiveness (culture, creativity, digital world, climate, energy and mobility . . . );<br>• An innovative Europe. |
| Next Generation EU (European Commission, 2020) [51] | Temporary recovery instrument<br>• More ecological Europe, digital, resilient and better adapted to present and future challenges;<br>• Research and investigation;<br>• Climate and digital transitions;<br>• Preparedness, recovery and resilience, through the Recovery and Resilience Mechanism, rescEU and a new health program, EUproHealth. |
| SURE Program (European Commission, 2020) [53] | • Temporary Support to mitigate Unemployment Risks in an emergency;<br>• Mobilization of financial resources to combat the social and economic impact of the coronavirus outbreak. |
| Mechanisms for a Fair Transition (European Commission, 2020) [58] | • State aid rules to boost green investment;<br>• Financing for a fair transition, the InvestEU program;<br>• Technical assistance. |
| European Agricultural Fund for Rural Development (EAFRD) (European Commission, 2020) [59] | • Promotion of agricultural combativeness;<br>• Sustainable management of natural resources and action against climate change;<br>• Balanced territorial development of rural economies and communities |
| InvestEU (European Parliament, 2021) [55] | • Sustainable infrastructure;<br>• Research, innovation and digitalisation;<br>• Small and medium-sized businesses;<br>• Social investment and skills. Financing of projects in education, social housing training, schools, universities, hospitals, health care, long-term care and accessibility, social entrepreneurship, integration of migrants, refugees and vulnerable people. |
| Digital Europe (European Commission, 2021) [57] | • Advanced digital skills;<br>• Generalization of the use of digital technologies in all sectors of the economy and society. |

Source: Information extracted from the different funds and instruments.

The Skills Agenda [49] to which we have alluded is remarkable because it aspires to incorporate in Europe the need to use educational tools, both formally and informally, to guarantee the economic, social and environmental sustainability of future generations and thus emerge from the current crisis stronger and more resilient. From this approach, we think that it is essential to direct education towards the development of advanced cognitive skills such as critical thinking or problem solving in a generalized and systematic way, but also towards soft skills such as autonomy, creativity, emotional control, resilient capacity or communication skills. In short, content education is necessary but linked to the training of children, youth and adults in a complex set of knowledge, skills, abilities, attitudes and values in different contexts and that allows them to solve situations related to environmental policies and transform or maintain reality with sustainability criteria.

The EU approach requires the development of a flexible educational strategy, with cross-cutting and specific content that lends itself to the development of sustainability and resilience. Education for Sustainable Development must be conceived as an integral part of a quality education inherent in the concept of lifelong learning, and in both formal and non-formal education and informal education [60]. In this context and supported by international recommendations, we propose to promote in educational institutions activities immersed in the curriculum and extracurricular activities, from companies, associations, NGOs with the aim of linking formal, informal and non-formal education and combining technology and learning for sustainability and resilience. Mutual learning opportunities between society and the school community in given contexts are particularly valuable as they provide an opportunity to foster sustainability and resilience and an opportunity to prepare our young people as proactive citizens for a globalized world with multiple potential risks.

The complete vision of risk in the face of an approach in which it is considered something specific requires the collaboration and action of multiple agents, including educators that make up a training network that develops resilience as an educational competence. The adoption of resilience in education, outside its traditional work framework, provides us with a change of mentality that allows us to go from thinking about difficulties to thinking about possibilities and solutions from a holistic ecosystem approach. This new approach, in which education at all its levels plays a fundamental role, entails developing in the student several skills such as those we have mentioned (emotional control, communication skills, culture of effort, commitment, creativity...) and skills advanced cognitive skills such as critical thinking or problem solving.

Education constitutes the natural context for learning and the development of resilient capacity that must be learned as the individual evolves and entails integrating cognitive, affective, social and behavioural aspects through modelling and conditioning through the different agents of the educational community (parents, teachers, peers).

## 6. LOMLOE, an Opportunity to Include Sustainability and Resilience in the Spanish Educational System

International recommendations on sustainable education have been incorporated into the Spanish educational system with some delay. The first Organic Law that addresses these issues is the Law of General Organization of the Educational System (LOGSE) [61] from the perspective of "training in respect and defence of the environment" (Article 2); the "conservation of nature and the environment" (Article 13) to which is added in (Article 19) the importance of "Critically assessing social habits related to health, consumption and the environment".

To this law is added the Law of Quality of Education (LOCE) of 2002 [62] that addresses issues related to the environment in its article 34 and points out the need to "strengthen sensitivity and respect for the environment".

The third law that does make explicit references to sustainability is the Organic Law of Education (LOE) [63] which in article 1 includes "education for the ecological transition with criteria of social justice as a contribution to sustainability, environmental, social and economic". Likewise, it also makes reference in article 25 to education for sustainability in a transversal manner and in article 110, which refers to "Accessibility, sustainability and relations with the environment".

The recently approved LOMLOE in 2020 [64] includes, for the first time, innovative aspects of Education for Sustainable Development and Education linked to Global Citizenship. The first key aspect that this law introduces is that educational centers become "dynamic environments where Education for Sustainability that permeates learning" and "spaces for the custody and care of the environment" [65]. Since this change, it is established that the educational administrations, in coordination with the institutions and organizations in their environment, will promote the sustainability of the centers, their relationship with the natural environment and their adaptation to the consequences of climate change.

The second innovative aspect is that the law is committed to promoting safe, sustainable and healthy mobility in schools, focusing on active travel, ensuring safe school paths and promoting sustainable travel in different territorial areas as a source of experience. and vital learning. In summary, the Law aims towards an integration and mainstreaming of environmental education and for sustainability through the complementarity of formal and non-formal education with the purpose that it contributes to the acquisition of competencies for Sustainable Development and Education for World Citizenship as reflected in the 2030 Agenda [19].

The third noteworthy aspect of the law is that it establishes content on sustainability at each stage (Table 3). Our proposal is to incorporate resilience into these contents through the Royal Decrees that regulate the Law.

**Table 3.** Sustainability in the various educational stages inside the LOMLOE.

| Education Level | Main Contents |
|---|---|
| Primary Education (Third Cycle) "Education in Civic and Ethical Values" | Human rights and infancy.<br>• Sustainable development and world citizenship;<br>• Equality between men and women;<br>• Respect for diversity, promoting the critical spirit, the culture of peace and no violence and respect for the environment and animals;<br>• Responsible consumption;<br>• Health Education. |
| Secondary Education | Transversely.<br>• Health Education;<br>• Sustainability education and mutual respect;<br>• Cooperation between equals;<br>• "Education in civic and ethical values";<br>• World Citizenship;<br>• Education for Sustainable Development. |
| High School | Critical spirit.<br>• No discrimination of order, sex, religion or other circumstance;<br>• Physical and mental well-being;<br>• Safe and healthy mobility;<br>• Responsible attitude in the fight against climate change and in defence of sustainable development. |
| Professional Training | • Competences related to the commitment to sustainable development and the prevention of occupational and environmental risks. |
| University Grades | • Social collaboration and citizen commitment;<br>• Sustainable development and climate change;<br>• Environmental crisis, of health or economies;<br>• Health and healthy habits. |

Source. Own elaboration from the text of the LOMLOE [64].

## 7. The Inclusion of Resilience as an Element of Sustainable Dimension in the LOMLOE Curriculum in the European Framework

European Union policies have been using the concept of resilience for a long time, although, as we have explained throughout the article, the concept was initially associated with environmental policy to later give way to incorporation into Regulation (EU) 2021/241 of the European Parliament and of the Council [66], in which explicit mention is made of the need to apply "policies for the next generation, children and youth, such as education and skills development" (p. 19) in order to achieve recovery and strengthen resilience.

In line with the EU recommendations for the incorporation of sustainability in all areas, the new Spanish education law LOMLOE includes references to education for sustainability

and the SDGs, such as: "The importance of attending to sustainable development in accordance with the provisions of the 2030 Agenda" [64].

However, in the text of the LOMLOE, the word resilience does not appear. The word risk appears four times without referring to DRR. The Law only refers to environmental risks (the rest talks about ICT, risk of poverty and social risk exclusion, low school enrolment, occupational risks and professional responsibility) but not linked to disaster risk.

LOMLOE should also incorporate European recommendations [67] and adapt education for sustainability and resilience from an approach that encourages proactive implementation. From this approach, we refer to autonomy, creativity and emotional control and to the inclusion of advanced cognitive aspects such as critical thinking or problem solving. In this sense, it is necessary to educate in the complex and integrated set of knowledge, skills, abilities, attitudes and values that people develop in different contexts (social, educational, work, family) to solve situations related to environmental problems, as well how to operate and transform reality with sustainability criteria [68].

Our proposal for the development of LOMLOE is to include resilience in education from a socio-ecological point of view, as indicated by Escalera and Ruiz [69], stating that resilience "can be understood as the capacity of a socio-ecosystem subject to some kind of stress—in the most basic sense of the term—or profound change—not necessarily negative—to regenerate itself without substantially altering its form and functions, in a kind of creative conservation "(p. 111, in quotation marks in the original).

The educational reform of the LOMLOE offers an opportunity to create a new educational model with two aspects; on the one hand, the acquisition of competences for the training and personal development of teachers [31], and on the other, their didactic capacity for the training of students in the competencies that enhance sustainability and resilient capacity.

Introducing resilience as an educational competence provides us with a change in mentality that allows us to go from thinking about difficulties to thinking about possibilities and solutions from a holistic ecosystem approach [70]. For this, it is necessary to develop emotional control, communication skills, a culture of effort, commitment, creativity or solving the problems to which we have alluded to.

We agree with Gónzalez and Meira [71] that education for sustainability focused on climate change implies preparing for the disaster, mitigating its effects locally and globally, adapting to the consequences and making informed decisions about the current situation. To do this, approaches, methodologies and tools are required that are not yet in our educational system in a generalized way. However, these approaches and methods appear to be incorporated into the LOMLOE considering sustainability and recognizing sustainable development. The results will be seen once it has been definitively implemented.

Additionally, it is a commitment assumed at an international level. On 20 May 2021, Spain passed a Climate Change and Energy Transition Law [72] to help achieve climate neutrality by 2050. In its title VIII, the law addresses education and training for development sustainability and caring for the climate through resilient societies. It also highlights the promotion of research, development and innovation to respond to climate change and promote the energy transition. In addition, in title VIII, considering citizen participation, the law contemplates the creation of a citizen assembly in which solutions to global warming will be discussed, advised by a committee of experts.

## 8. Discussion

Natural or man-made disasters and social changes are more and more numerous and are it is necessary to design strategies to train more resilient and proactive people. The task is complex and will become more widespread in all areas of the planet. It is in this context that the EU is proposed to require deep and systematic transformations, but education is postulated as a bulwark to train in skills that develop resilience and work towards sustainability. The concepts of sustainability and resilience go hand in hand in fields such as environmental management, urban planning, agriculture or the workplace,

but we must go one step further and, in line with the regulations generated by the EU, configure an educational system where, in a transversal and specific way, contents are worked on that develop sustainability but also resilient capacity. All of this occurs through strategies, processes and practices that support the transition to reforms that have already been taking place in other fields for some years.

In the 1960s, international concern for the environment began, which, after several summits has become the current commitment to the concept of sustainability in education within the framework of the SDGs [18], among which the goal 4.7 stands out, which is committed to every student acquiring the theoretical and practical knowledge necessary to promote sustainable development, including through education for sustainable development and sustainable lifestyles. The recent LOMLOE [64] organic law passed in 2020 is correctly framed by international recommendations.

However, in the first third of the 21st century, the COVID-19 crisis has revealed that, in order to achieve sustainable development, it is necessary for communities to strengthen their resilient capacity to face and overcome new challenges such as environmental disasters and pandemics. This requires that the education system assume new functions such as containment and that of establishing links between formal and non-formal education for the creation not only of individuals but also of resilient communities.

This article provides the need to link education for sustainability with the resilience factor. Although since the 1990s education has been recognized as a fundamental element to combat climate change and move towards a sustainable society, resilience and education for disaster risk has been addressed almost exclusively in the summits held on Japanese territory. In other words, on the one hand, the World Summit on Sustainable Development organized by the United Nations in 2002 in Johannesburg [17] declared the Decade of Education for Sustainable Development, that is, an education for sustainability and not about sustainability. On the other hand, in 2005 in Hyogo, an Action Plan was approved that considers education for a new action plan between 2005 and 2015 [40].

This line of separation of the contents between sustainability and resilience has been maintained until the present time since, in the first third of the 21st century, two transcendental documents on the subject were again approved separately in 2015. On the one hand, the 2030 Agenda for Sustainable Development and, on the other hand, the Sendai Framework for Disaster Risk Reduction (2015–2030) [43].

However, as mentioned by Belmonte-Ureña et al. [73], critical voices report that sustainable development considering the limitation of the planet's resources is impossible. Therefore, they propose degrowth as a theoretical-practical framework. Thus, degrowth comes to provide a transformative vision and to offer a radical change in the direction of a new form of social and economic organization.

The COVID-19 crisis has been a clear turning point in education for sustainability and resilience as it has been revealed that to achieve sustainable development it is necessary for communities to strengthen their resilient capacity to face and overcome new sustainable challenges for an education that responds to the two factors claimed.

We can speak of the fact that the European bet has focused on the value of resistance and not resilience. Since the early 1950s, it has focused primarily on economic and financial issues, which means that resilience maintains that bias. However, the EU, mainly as of 2015, has incorporated the sustainable and resilient dimension around the definition of competencies but disconnected from the conception of the field of natural disasters.

A clear example of the approach to resilience linked to the professional field is the 2016 European Skills Agenda [49], which was renewed in 2020 in order to train more resilient and proactive people in the professional field. This synchronization of working life with the development of specific and transversal skills such as cooperation, critical thinking, media, environmental and health literacy, civic skills or resilience leads to the modernization of education, transforming this into a more attractive and flexible action in the face of the digital age and the ecological transition.

Climate change adaptation, disaster risk reduction and sustainable development are intrinsically interrelated and interdependent areas that converge on the importance of resilience. We also know that potential disasters and social changes are more and more numerous and are more widespread in all areas of the planet.

Bearing in mind the lessons learned after COVID-19 and after an in-depth analysis of international guidelines for both sustainable development and resilient education, we can contextualize the Spanish LOMLOE [64] Law approved in 2020 as an opportunity to link adaptation to climate change, disaster risk reduction and sustainable development as intrinsically interrelated and interdependent fields in which resilience played a fundamental role. The current debate on the Royal Decrees that they are going to develop constitutes a unique opportunity to enrich the broad proposal of content on sustainability and resilience that has been proposed in said law.

Learning processes related to building resilient capacities to achieve threat and vulnerability reduction could promote sustainable development. However, integrating policies that foster individual and social resilience through sustainable development remains a challenge.

**Author Contributions:** All authors were involved in conceptualization, research, formal analysis and writing. All authors have contributed to the development of this paper. Review and editing were done by E.G.-S., J.P.-H., I.N.-G. and P.-T.E.-G. All authors have read and agreed to the published version of the manuscript.

**Funding:** This research was funded by the Universidad Nacional de Educación a Distancia (UNED), Spain. The APC was funded by grant program "Cooperación Universitaria para el Desarrollo para la consecución de los Objetivos de Desarrollo Sostenible (ODS) UNED 2020". Project title: Study of national strategies to deal with the COVID-19 crisis: the cases of South Korea and Japan (EENACC).

**Institutional Review Board Statement:** Not applicable.

**Informed Consent Statement:** Not applicable.

**Data Availability Statement:** The data that support the findings of this study are available from the corresponding author, upon reasonable request.

**Acknowledgments:** The authors sincerely thank the valuable support of the Vice-Rectorate for Internationalization of the UNED.

**Conflicts of Interest:** The authors declare no conflict of interest. The funders had no role in the design of the study; in the collection, analyses, or interpretation of data; in the writing of the manuscript, or in the decision to publish the results.

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
