# Peer review of "The Inclusion of Resilience as an Element of the Sustainable Dimension in the LOMLOE Curriculum in a European Framework"

_sustainability, doi:10.3390/su132413714_

Round 1

Reviewer 1 Report

The topic of the paper is veryimportant and timely. Unfortunately, it lacks a theoretical framwork and sufficent methodologcal and methodical explications. Hence, the paper appears as a report, not as a research-paper. The major notions and concepts are being used must be defined and reflected (e. g. sustainability, desaster education, resilience). There is a body of research that discusses these conetsted termes broadly and critically; that has to b taken into account. Otherwise educational questions cannot be addressed: it is not self-evident that resilience is an adequat answer to desasters resulting from climate change, it is a controverse.

Author Response

After reading the review for the article we have revised the manuscript following your suggestions.

Your suggestions that more theoretical background should be provided have been fulfilled. We have backed are arguments with updated academic references concerning the link between sustainability and resilience which is one of the main statements of our research.  

Following the reviewer recommendations we have included the definitions of sustainability, disaster reduction education and resilience. We explore the link between them , the differences and the relevance in todays international recommendations.

The article provide the academic auditorium with a review of the international recommendations concerning sustainability and resilience and show how the national education systems should incorporate the international recommendations. The selected national education system is the Spanish one which after the COVID_19 pandemia approved an Organic Law in 2020 where sustainability and global citizenship are a pillar of the reform. However from our point of view we argue that this is not enough and education for building resilient individuals and communities should be considered.

In what concerns the methodology we have applied the comparative methodology typical of the European tradition which is based on humanistic sciences and in particular, analysing from a historical approach. From this prospective we  offered the analysis of a national education system reform, such as the 2020,  and we criticise the lack of the inclusion of the resilience dimension. This methodology is the typical of the comparative education method in the European Tradition.

Reviewer 2 Report

The paper is well-written, comprehensive and makes an adequate reflection. The tables are clarifying and useful for further research as well as for generating transfer to public sectors that use indicators associated with the object of study. In particular, Table 3 can be used by active teachers in schools. 
It is recommended to simplify the information in the tables because they have become extensive. If other columns were incorporated in table 2, the information would be more systematized.

Author Response

Thank you for your suggestions

Following the reviewer´s recommendations we have simplified the information in the tables so they are no longer so extensive. In addition we have included another column so that the information is more systhematised.

Reviewer 3 Report

A very good paper.

Author Response

Thank you very much for your suggestions

Round 2

Reviewer 1 Report

The article has been improved significantly.